# Potential Cutaneous Applications of *Boesenbergia rotunda* Extract Based on Its In Vitro Anti-Melanogenic and Anti-Fibroproliferative Properties

**DOI:** 10.3390/ijms26094319

**Published:** 2025-05-01

**Authors:** Nuntida Salakshna, Wilai Thanasarnaksorn, Phongthon Kanjanasirirat, Kedchin Jearawuttanakul, Napason Chabang, Noppawan Rangkansenee, Uraiwan Panich, Saowalak Thanachaiphiwat, Suparerk Borwornpinyo, Suradej Hongeng

**Affiliations:** 1Samitivej Esthetics Institute, Samitivej Sukhumvit Hospital, Bangkok 10110, Thailand; nppn99@hotmail.com; 2Science Division, Mahidol University International College, Nakhon Pathom 73170, Thailand; 3Division of Dermatology, Department of Medicine, Faculty of Medicine, Ramathibodi Hospital, Mahidol University, Bangkok 10400, Thailand; 4Division of Dermatology, Chulabhorn International College of Medicine, Thammasat University, Pathum Thani 12120, Thailand; 5Department of Pathobiology, Faculty of Science, Mahidol University, Bangkok 10400, Thailand; phongthon.kan@mahidol.ac.th; 6Excellent Center for Drug Discovery (ECDD), Faculty of Science, Mahidol University, Bangkok 10400, Thailand; o-nion@hotmail.com (K.J.); bsuparerk@gmail.com (S.B.); suradej.hon@mahidol.ac.th (S.H.); 7School of Bioinnovation and Bio-Based Product Intelligence, Faculty of Science, Mahidol University, Bangkok 10400, Thailand; napason.cha@mahidol.ac.th; 8Institute of Molecular Biosciences, Mahidol University, Nakorn Pathom 73170, Thailand; noppawan.ran@mahidol.ac.th; 9Department of Pharmacology, Faculty of Medicine, Siriraj Hospital, Mahidol University, Bangkok 10700, Thailand; uraiwan.pan@mahidol.ac.th (U.P.); nummthana@gmail.com (S.T.); 10Department of Biotechnology, Faculty of Science, Mahidol University, Bangkok 10400, Thailand; 11Department of Pediatrics, Faculty of Medicine, Ramathibodi Hospital, Mahidol University, Bangkok 10400, Thailand

**Keywords:** *Boesenbergia rotunda*, panduratin A, fingerroot, anti-melanogenic, anti-fibroproliferative

## Abstract

The rhizomes of *Boesenbergia rotunda* (fingerroot) and its bioactive compound, panduratin A, exhibit potent anti-inflammatory, antioxidant, and anti-proliferative properties. The aim of this study was to investigate the anti-melanogenic (including anti-tyrosinase) effects of *B. rotunda* extract and panduratin A in B16F10 melanoma cells induced by UVA radiation. The effects of the compounds on fibroblast proliferation and migration were also assessed by scratch wound healing assays in human primary fibroblasts. The results showed that *B. rotunda* extract and panduratin A significantly reduced tyrosinase activity and cellular melanogenesis induced by UVA radiation in a dose-dependent manner. The compounds also demonstrated inhibitory effects on fibroblast cell migration and proliferation. These findings suggest that *B. rotunda* extract may have potential therapeutic effects on skin hyperpigmentation and fibroproliferative skin disorders.

## 1. Introduction

Hyperpigmentation related to sun exposure (such as melasma and post-inflammatory hyperpigmentation (PIH)) is one of the most prevalent dermatological concerns, with challenging treatment strategies [1,2]. Ultraviolet radiation has been identified as the significant contributor to this problem, especially UVA, which humans are most exposed to [3]. While the exact mechanism of skin hyperpigmentation is still unknown, it involves increased melanocyte activity [3]. Additionally, fibroproliferative skin disorders related to wound healing such as hypertrophic scars and keloids are another substantial cosmetic concern. These conditions are characterized by thick, overgrown, and raised scars caused by chronic inflammation and fibroblast proliferation [4,5]. This clinically results in pain, itchiness, limited mobility, and a negative impact on quality of life [6,7,8]. Given these challenges, there is growing interest in exploring natural agents that can mitigate cellular melanogenesis and inhibit fibroblast proliferation as complementary strategies for preventing skin hyperpigmentation and hypertrophic scar or keloid formation [9,10]. *Boesenbergia rotunda* (English: fingerroot, Thai: krachai) is primarily cultivated in tropical regions such as Thailand and Indonesia [11]. The rhizome of *B. rotunda* extract and its bioactive flavonoid, panduratin A (chemical structure was shown in Figure 1), have shown various pharmacological effects including potent anti-inflammatory, antioxidant, and anti-proliferative properties [12,13,14].

UVA radiation, which penetrates deeper into the dermis than UVB, plays a significant role in photoaging and hyperpigmentation. While traditionally associated with immediate pigment darkening via melanin oxidation, recent studies suggest that UVA also contributes to de novo melanogenesis through oxidative stress-related mechanisms [15,16,17]. UVA-induced ROS formation leads to redox imbalance and oxidative DNA damage in melanocytes, which in turn can activate melanogenic enzymes such as tyrosinase, independent of classical UVB-induced p53 signaling. In this context, the antioxidant transcription factor nuclear factor E2-related factor 2 (Nrf2) plays a critical role in mitigating UVA-mediated oxidative stress and maintaining cellular homeostasis in melanocytes and keratinocytes [18].

The novelty of our study lies in specifically addressing UVA-induced melanogenesis, an area less well characterized than UVB-mediated pigmentation pathways.

This study aimed to investigate the effect of *B. rotunda* extract and panduratin A on cellular melanogenesis induced by UV radiation using B16F10 mouse melanoma cells and the effects of those compounds on fibroblast proliferation and migration using scratch wound healing assays.

## 2. Results and Discussion

### 2.1. In Vitro Study of the Effect of Panduratin A and B. rotunda Extract on Cellular Melanogenesis Induced by UV Radiation

#### Cytotoxicity

In the cytotoxicity test of phenolics in B16F10 cells, all concentrations of panduratin A (0.625–2.5 μM) or crude extract (0.625–10 μg/mL) did not cause significant cell death (Figure 2a,b).

### 2.2. Panduratin A and Fingerroot Extract Inhibited UVA-Induced Melanin Content in B16F10 Cells

Panduratin A at a concentration of 2.5 μM and crude extract at a concentration of 5 and 10 μg/mL significantly inhibited the melanin content in UVA-irradiated cells in a dose-dependent manner. The inhibitory effect of panduratin A at 2.5 μM was 28.4% (* *p* = 0.0339), and the inhibitory effects of crude extract at 5 and 10 μg/mL were 41.6% (** *p* = 0.0187) and 55.4% (*** *p* < 0.001), respectively, without significant difference from the positive control (Figure 2c–e).

### 2.3. Panduratin A and Fingerroot Extract Inhibited UVA-Induced Tyrosinase Activity in B16F10 Cells

The tyrosinase activity assay showed that panduratin A at the concentration of 1.25 and 2.5 μM and crude extract at the concentration of 2.5, 5, and 10 μg/mL significantly inhibited tyrosinase enzyme in UVA-irradiated cells in a dose-dependent manner. The inhibitory effects of panduratin A at 1.25 and 2.5 μM were 39.2% (** *p* = 0.0092) and 54% (*** *p* < 0.001) respectively, and the inhibitory effects of crude extract at 2.5, 5, and 10 μg/mL were 30.5% (** *p* = 0.0092), 43.7% (*** *p* < 0.001), and 46.4% (*** *p* < 0.001), respectively, with more inhibitory effects than the positive control (Figure 2f–h).

### 2.4. Summary of Study in Cellular Melanogenesis

The melanin content and tyrosinase activity of UVA-irradiated B16F10 cells was significantly reduced by panduratin A and *B. rotunda* extract in a dose-dependent manner without causing cytotoxicity.

### 2.5. Evaluation of the Wound-Healing Effect

FBS, the positive control, showed activity that promoted wound closure. On day 2, the wound region using 10% FBS was significantly smaller than 0% FBS (1.3 × 10^6^ vs. 1.9 × 10^6^ µm^2^, *p* = 0.0196, respectively). Likewise, the wound region using 20% FBS on day 2 was significantly smaller than 0% FBS (1.0 × 10^5^ vs. 1.9 × 10^6^ µm^2^, *p* = 0.0174, respectively) (Figure 3).

Contrasting with FBS, both of our tested compounds, *B. rotunda* extract (fingerroot extract) and panduratin A, affected the inhibition of fibroblast migration and proliferation. On day 1, the wound regions with fingerroot extract and panduratin A were significantly larger compared to 0.1% DMSO (2.5 × 10^6^ and 3.7 × 10^6^ vs. 1.0 × 10^6^ µm^2^, *p* = 0.0325 and *p* = 0.0013, respectively). Similarly, the wound regions with fingerroot extract and panduratin A were significantly larger compared to 0.1% DMSO on day 2 (3.1 × 10^6^ and 4.8 × 10^6^ vs. 1.3 × 10^5^ µm^2^, *p* = 0.0004 and *p* < 0.0001, respectively) (Figure 4).

Moreover, fingerroot extract inhibited scratch closure in fibroblast cells in a dose-dependent manner from 0 to 20 µg/mL over 24 h. At 24 h, it was observed that the wound regions of 10 and 20 µg/mL fingerroot extract were significantly larger than the wound region of 0 µg/mL fingerroot extract (4.2 × 10^6^ and 6.3 × 10 µm^2^ vs. 1.8 × 10^6^ µm^2^, *p* < 0.0001 and <0.0001, respectively) (Figure 5). Similarly, panduratin A inhibited wound closure in a dose-dependent manner. At 24 h, the wound regions of 2.5, 5, 10, 20 µM of panduratin A were significantly larger than the wound region of 0 µM (3.6 × 10^6^, 5.6 × 10^6^, 6.5 × 10^6^, 7.5 × 10^6^ vs. 1.8 × 10^6^ µm^2^, *p* = 0.0011, *p* < 0.0001, *p* < 0.0001, *p* < 0.0001, respectively) (Figure 6). This shows that panduratin A exhibits more effective results against fibroblast migration and proliferation at lower concentrations than fingerroot extract.

*B. rotunda*, commonly known as fingerroot, is a herbaceous plant that belongs to the Zingiberaceae family or the ginger family [11]. It contains the bioactive compound, panduratin A, a cyclohexenyl chalcone derivative belonging to the flavonoid family [14]. *B. rotunda* and panduratin A have been investigated for their potential therapeutic effects of antioxidation, anti-proliferation, and anti-inflammation [19,20]. Skin hyperpigmentation and fibroproliferative skin disorders related to wound healing, including keloids and hypertrophic scars, significantly impact quality of life and cosmetic appearance [21,22]. The dual inhibitory effects of *B. rotunda* and panduratin A on these processes offer promising approaches for the development of novel skincare and wound-healing interventions. This study examines the current understanding of the effect of *B. rotunda* and panduratin A on cellular melanogenesis induced by UVA radiation and on fibroblast proliferation.

Ultraviolet (UV) radiation is a well-established environmental factor contributing to the pathogenesis of skin hyperpigmentation. Melanogenesis, the biological process through which melanin is synthesized to protect the skin from UV-induced damage, can become dysregulated, resulting in pigmentary disorders such as melasma and post-inflammatory hyperpigmentation [22].

In the present study, *B. rotunda* extract and its major bioactive constituent, panduratin A, were found to significantly reduce melanin content and inhibit tyrosinase activity—the rate-limiting enzyme in melanogenesis—in UVA-irradiated B16F10 melanoma cells. These effects occurred in a dose-dependent manner and without evidence of cytotoxicity. Our findings corroborate previous research demonstrating the capacity of panduratin A to suppress melanin synthesis, inhibit tyrosinase activity, and down-regulate the expression of melanogenesis-associated genes, including tyrosinase-related protein 1 (TRP-1) and tyrosinase-related protein 2 (TRP-2) [23]. The current study is, to our knowledge, the first to report these effects in a UVA-induced cellular melanogenesis model.

Panduratin A is a cyclohexenyl chalcone derivative, a structural characteristic that may underlie its strong inhibitory effect on tyrosinase activity [23]. In addition to its antimelanogenic properties, panduratin A is known for its anti-inflammatory, antioxidant, and antimicrobial activities, which have attracted attention for potential therapeutic applications [24,25,26,27,28]. Likewise, *B. rotunda* exhibits significant antioxidant and anti-inflammatory properties. Previous studies have shown that *B. rotunda* enhances the cellular antioxidant defense system while simultaneously suppressing lipid peroxidation and modulating key signaling pathways, including protein kinase B (Akt) and the nuclear factor kappa-light-chain-enhancer of activated B cells (NF-κB). These actions suggest that the photoprotective effects of *B. rotunda* in UV-induced melanogenesis may be attributed, at least in part, to its antioxidant mechanisms [29,30].

Moreover, dysregulation of the antioxidant transcription factor Nrf2 has been shown to exacerbate UVA-induced melanogenesis, highlighting the redox-sensitive nature of this process [18]. This suggests that, although UVA may not initiate melanogenic gene expression through the classical UVB-mediated pathways, it can still promote melanogenesis by inducing oxidative stress and disrupting redox signaling [31,32,33].

These findings are consistent with growing evidence that UVA radiation can promote melanogenesis through oxidative stress mechanisms, independent of its immediate pigment-darkening effects. Studies have demonstrated that UVA exposure leads to the accumulation of reactive oxygen species (ROS), oxidative damage, and depletion of intracellular glutathione, all of which are associated with increased tyrosinase activity and melanin content in melanocytes and melanoma cells [15,16,17,18,33], Notably, knockdown of nuclear factor erythroid 2–related factor 2 (Nrf2), a master regulator of cellular antioxidant responses, was shown to exacerbate UVA-induced melanogenic effects, implicating oxidative stress—rather than direct DNA damage—as a principal mediator of UVA-driven pigmentation. Moreover, these redox-sensitive responses are modulated by upstream mitogen-activated protein kinases (MAPKs), including ERK, JNK, and p38, which regulate the nuclear translocation and activity of Nrf2 [18,33].

Additionally, Esposito et al. (2022) found that compared to UVB, UVA causes significantly less erythema but is more potent in triggering pigment darkening—both immediate and persistent—as well as delayed tanning, particularly in individuals with darker skin tones [3,34]. Unlike UVB, UVA does not directly impact key skin biomolecules. Instead, it acts indirectly by transferring energy to chromophores, which then produce reactive species that cause oxidative stress [35]. Moreover, long-wavelength UVA and visible light (VL) can work together to enhance skin pigmentation and erythema [36]. During typical daily activities, people are most commonly exposed to UVA and VL. However, current commercial sunscreens do not offer complete protection in this spectrum.

Taken together, our findings suggest that *B. rotunda* extract and its bioactive compound, panduratin A, may help reduce UVA-induced melanogenesis, as demonstrated by decreased melanin content and tyrosinase activity in treated cells. While the detailed mechanisms remain to be fully explored, these results support the potential of *B. rotunda* as a natural candidate for preventing UVA-related skin pigmentation.

In addition to its inhibitory effects on melanogenesis, our study showed that *B. rotunda* and panduratin A inhibit fibroblast proliferation and migration in a dose-dependent manner. Fibroblast proliferation is essential for wound healing and tissue regeneration, and thereby these compounds offer potential therapeutic benefits in conditions characterized by excessive tissue scarring or fibrosis, such as hypertrophic scars and keloids.

*B. rotunda* and panduratin A are known to have potent anti-proliferative, antioxidant, and anti-inflammatory effects [12,13,14]. These key features are the basis of anti-keloid treatment. Several studies indicate that panduratin A inhibits the release of many cytokines involved in the wound healing process and keloid formation. A study showed that panduratin A significantly decreased the level of the fibrinogenesis markers TGF-β1 and PDGF in a dose-dependent manner, producing slower fibroblast cell proliferation and migration in an in vivo rat model experiencing liver cirrhosis [37]. Also, a previous study demonstrated that panduratin A substantially down-regulated MMP-2 and TIMP-1, which are important regulators of extracellular matrix degradation and remodeling and are highly expressed in keloid fibroblasts. Additionally, it showed that panduratin A reduced oxidative stress in liver fibrosis [38]. Moreover, panduratin A inhibits many inflammatory mediators including nitric oxide and prostaglandin E2 (PGE_2_). The excess collagen deposition in keloid pathology is also related to the overproduction of nitric oxide [24,39]. The evidence found that among many other phytochemical compounds, panduratin A is a potent inhibitor of nitric oxide production. While PGE_2_ causes inflammation and promotes cell proliferation and angiogenesis, panduratin A exhibits strong inhibition of PGE_2_ [40]. These mechanisms support that panduratin A could be a potential substance for inhibiting the formation of keloids.

One limitation of our fibroblast model is the absence of pro-fibrotic cytokines such as TGF-β1, IL-6, and PDGF, which are commonly elevated in keloid and hypertrophic scarring. Future studies incorporating these key signaling molecules will be necessary to determine whether *B. rotunda* and panduratin A selectively inhibit pathological fibroblast over-proliferation without compromising normal wound healing. Additionally, while reduced fibroblast proliferation may be beneficial in fibrotic conditions, it also raises potential concerns regarding delayed wound repair. Therefore, careful dose optimization and in vivo validation will be critical to distinguish the desired anti-fibrotic effects from unintended impairment of physiological healing processes.

Our study demonstrated the potential cutaneous applications of *B. rotunda* extract and its bioactive compound, panduratin A, through their anti-melanogenic and anti-fibroproliferative properties. The dual inhibitory effects of *B. rotunda* on both melanogenesis and fibroblast proliferation hold promise for addressing a wide range of dermatological conditions, including hyperpigmentation disorders and hypertrophic scarring. However, further studies are needed to confirm these findings and determine the optimal concentration and mechanism of action of these substances.

## 3. Materials and Methods

The study utilized UVA-induced models in B16F10 melanoma cells to assess melanogenesis and scratch wound healing assays in human dermal fibroblasts to evaluate cellular migration.

### 3.1. Crude Extract and Panduratin A Preparation

*B. rotunda* rhizomes (Fingerroot) and panduratin A were received from the Excellent Center for Drug Discovery (ECDD) laboratory (Faculty of Science, Mahidol University, Bangkok, Thailand) in a solid powder form. The known concentrations of both fingerroot extract and panduratin A stock solutions were made by dissolving the powder received in 0.1% dimethyl sulfoxide (DMSO). Thereafter, the stock solutions were diluted to the desired concentrations used in the assays.

### 3.2. Cell Culture

#### 3.2.1. B16F10 Mouse Melanoma Cells

The B16F10 mouse melanoma cells (ATCC, Manassas, VA, USA) were grown in Dulbecco’s modified Eagle medium (DMEM) that contained 10% fetal bovine serum (FBS) and 1% penicillin (100 units/mL)/streptomycin (100 μg/mL). The cells were kept at a temperature of 37 °C in a humidified incubator containing 5% CO_2_/95% air.

#### 3.2.2. Human Primary Fibroblasts

Human primary fibroblasts were purchased from American Type Culture Collection (ATCC) (Manassas, VA, USA). A hemocytometer was used for cell counting. The cells were grown in a 96-well plate with Fibroblast Growth Medium 2 (PromoCell, Heidelberg, Germany), followed by incubation in a humidified CO_2_ incubator at 37 Celsius, 5% CO_2_.

#### 3.2.3. Cytotoxicity Test of Panduratin A and *B. rotunda* Extract on B16F10 Cells

Cytotoxicity was determined using MTT (3-(4, 5-dimethylthiazol-2-yl)-2, 5-diphenyltetrazolium bromide) assay. This colorimetric measurement measured the reduction of yellow MTT to purple formazan using mitochondrial succinate dehydrogenase in living cells to evaluate cell viability. The cells were exposed to test phenolics for 24 h with concentrations of up to 80 μM panduratin A and 80 μg/mL crude extract. After washing the cells with PBS, the plates were incubated for 1 h at 37 °C with a medium containing 0.2 mg/mL of MTT in a 5% CO_2_ incubator. Then, 200 µL of DMSO was used to solubilize the formazan crystals in each well. The plates were shaken gently for 10 min. Afterwards, the optical density was measured by a spectrophotometer at 595 nm. Viable cells were expressed as a percentage of the control (100%, non-UVA-irradiated and non-treated cells).

#### 3.2.4. Treatment of Cells with Phenolics and UVA Irradiation

The cells were exposed to panduratin A at selected concentrations of 0.312, 0.625, 1.25, and 2.5 μM or crude extract at selected concentrations of 1.25, 2.5, 5, and 10 μg/mL for 30 min in phosphate-buffered saline (PBS). After treatment with phenolics, cells were exposed to 8 J/cm^2^ UVA radiation (xenon arc lamp, with a wavelength range of 320–400 nm, Dermalight ultrA1). Subsequently, cells were harvested for measurement of melanin content and tyrosinase activity. Non-UVA-irradiated and non-phenolic-treated cells were used as the negative control. Arbutin 20 μM was used as the positive control.

#### 3.2.5. Preparation of Total Cell Lysate

The cells were harvested and concentrated through centrifugation before being lysed in an ice-cold extraction buffer. The buffer contained 50 mM Tris-HCl, 10 mM ethylenediaminetetraacetic acid (EDTA), 1% (*v*/*v*) Triton X-100, phenylmethylsulfonyl fluoride (PMSF) at a concentration of 100 mg/mL, pepstatin A at a concentration of 1 mg/mL in DMSO, and leupeptin at a concentration of 1 mg/mL in H_2_O, with a pH of 6.8. Following this, the cells were subjected to further centrifugation at 10,000 rpm for 10 min, and the complete cell lysate was collected.

#### 3.2.6. Melanin Content Assay

Cells were pre-treated with panduratin A (0.312–2.5 μM) or crude extract (1.25–10 μg/mL) in PBS for 30 min before exposure to UVA (8 J/cm^2^). The total melanin contents were then extracted by 200 µL of 1N NaOH at 60 °C for 1 h, and the lysate was collected in microtubes. Subsequently, 100 µL of lysis buffer (50 mM Trisma base; 10 mM EDTA; 1% (*v*/*v*) Triton X-100; 0.57 µM pepstatin A 2 µM leupeptin) was added to extract protein and subsequently centrifuged at 10,000 rpm, 4 °C for 10 min. The supernatant was collected by centrifugation, and 100 µL melanin lysate was added to a 96-well plate to be measured by spectrophotometry at 475 nm. The melanin content (μg/mg protein) was calculated by comparison to a standard curve derived using synthetic melanin of non-UVA-irradiated and non-phenolic-treated cells (100%).

#### 3.2.7. Tyrosinase Activity Assay

Cells were pre-treated with panduratin A (0.312–2.5 μM) or crude extract (1.25–10 μg/mL) in PBS for 30 min before exposure to UVA (8 J/cm^2^). Cellular tyrosinase activity was determined by measuring the rate of oxidation of L-DOPA. Cells were lysed with 100 µL of lysis buffer (0.1 M phosphate buffer pH 6.8, 0.1% Triton X-100) and incubated for 10 min at 4 °C with gentle shaking. The cell lysates were then clarified by centrifugation at 10,000 rpm for 10 min at 4 °C. The supernatant was collected and incubated with 10 µL of substrate solution (20 mM L-DOPA), and 90 µL of cell lysates was assayed on a 96-well plate at 37 °C. Subsequently, the DOPAchrome production rate—reflecting tyrosinase activity—was measured by spectrophotometry at 475 nm for 1 h at 37 °C. The results of tyrosinase activity (unit/mg protein) were calculated by comparison with a standard curve derived using purified mushroom tyrosinase (2034 units/mg) of non-UVA-irradiated and non-phenolic-treated cells (100%).

#### 3.2.8. Protein Quantification and Normalization

Protein concentrations were determined using the Bradford assay, which allowed for accurate quantification of total protein in each sample. These values were then used to normalize tyrosinase activity and melanin content, ensuring that results were expressed relative to protein content (mg) for consistency across samples.

#### 3.2.9. In Vitro Wound Healing, Proliferation, and Migration Assay

Human primary fibroblasts were cultured in 96 well plates at 20,000 cells/well with Fibroblast Growth Medium 2 and were incubated for 24 h in a humidified CO_2_ incubator at 37 Celsius, 5% CO_2_. After the cells formed a monolayer, wounds were generated using sterile 200 μL micropipette tips. Cells were washed with Dulbecco’s phosphate-buffered saline (DPBS) 3 times and were treated with 4 conditions: fetal bovine serum (FBS) as a positive control, DMSO, fingerroot extract, and panduratin A. FBS, a supplement serum for in vitro cell growth, acted as a positive control. DMSO is an organosulfur solvent that dissolves both polar and nonpolar compounds. As our tested compounds were to be dissolved in DMSO, the effect of DMSO alone needed to be determined first. FBS and DMSO were purchased from Merck (Schuchhardt, Darmstadt, Germany). Fingerroot extract and panduratin A were received from the ECDD laboratory (Faculty of Science, Mahidol University, Thailand). The movement of fibroblast cells was tracked with the Operetta^®^ high-content imaging system, (Perkin Elmer, Springfield, IL, USA) to determine the wound-healing process.

### 3.3. Statistical Analysis

The statistical significance of the differences between the phenolic-treated groups or UVA-irradiated groups and the control was evaluated by a one-way ANOVA followed by Dunnett’s multiple-comparison post-test using GraphPad Prism 5 statistical analytic software (GraphPad Software Inc., San Diego, CA, USA). The mean and standard deviation were adopted to express the data for each group. Statistically significant differences from basal or control values were classified as *, # *p* < 0.05; **, ## *p* < 0.01; and ***, ### *p* < 0.001.

The differences in the wound region of fibroblast migration and proliferation were analyzed using a two-way ANOVA. GraphPad Prism 9 software was used to determine the statistical significance and to construct the graphs. Statistical significance was defined as a value of *p* < 0.05.

## 4. Conclusions

This study investigated the potential of *B. rotunda*, a herb traditionally used for medicinal purposes, to prevent or mitigate cellular melanogenesis induced by UV radiation. The results showed that *B. rotunda* extract and its bioactive compound, panduratin A, reduced melanin production and tyrosinase activity in a dose-dependent manner, indicating its potential as a skin protectant against UV radiation. Moreover, they exhibited inhibition of fibroblast proliferation and migration. These findings suggest that *B. rotunda* may have promising applications in the development of natural skincare products with dual inhibitory effects on melanogenesis and fibroblast proliferation.

## Figures and Tables

**Figure 1 ijms-26-04319-f001:**
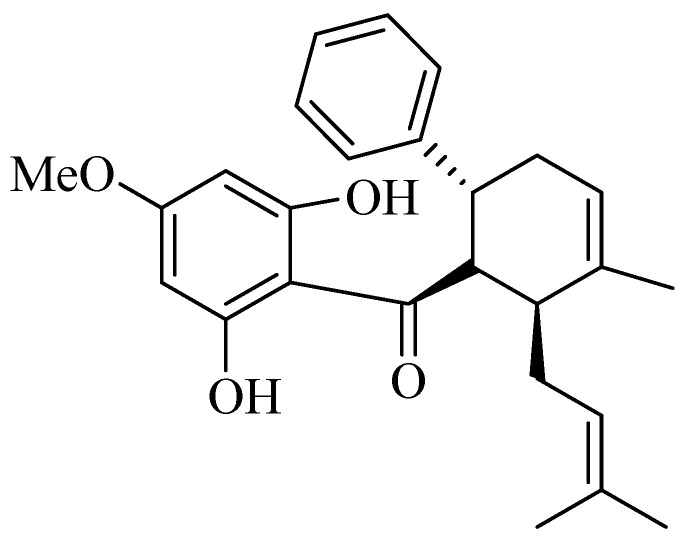
Chemical structure of panduratin A, the bioactive compound in *Boesenbergia rotunda*.

**Figure 2 ijms-26-04319-f002:**
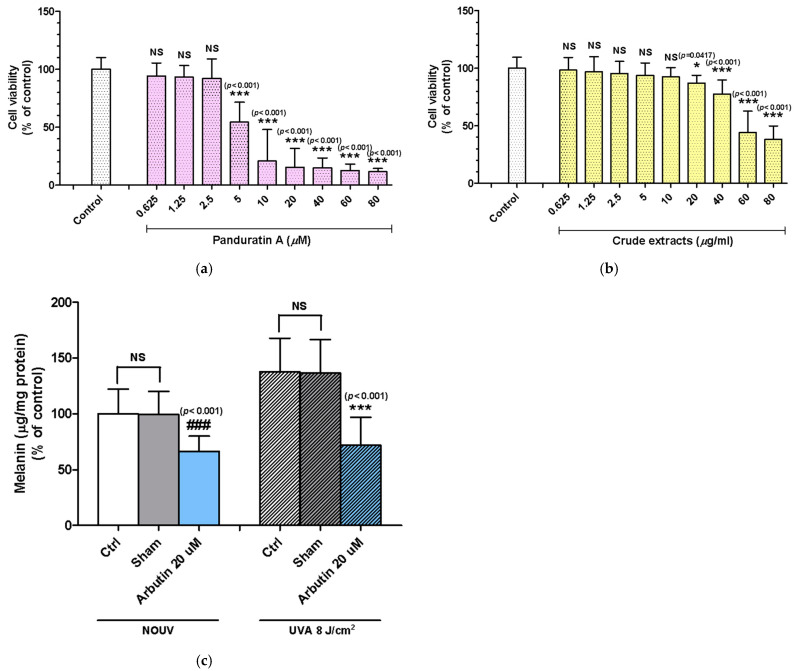
Effects of phenolics on cellular melanogenesis induced by UVA radiation. (**a**,**b**) Cytotoxicity test of phenolics in mouse melanoma (B16F10) cells. (**c**) UVA significantly induced melanin content in the control. (**d**) Effects of panduratin A (0.312–2.5 μM) and arbutin (20 μM) as a positive control on melanogenesis in B16F10 cells in response to UVA irradiation. (**e**) Effects of crude extract (1.25–10 μg/mL) and arbutin (20 μM) as a positive control on melanogenesis in B16F10 cells in response to UVA irradiation. (**f**) UVA significantly induced tyrosinase activity in the control. (**g**) Effects of panduratin A (0.312–2.5 μM) and arbutin (20 μM) as a positive control on tyrosinase activity in B16F10 cells in response to UVA irradiation. (**h**) Effects of crude extract (1.25–10 μg/mL) and arbutin (20 μM) as a positive control on tyrosinase activity in B16F10 cells in response to UVA irradiation. Data are expressed as mean (SD). The statistical significance of the differences was evaluated using a one-way ANOVA followed by Dunnett’s test. * *p* < 0.05; **, **##**
*p* < 0.01; ***, **###**
*p* < 0.001.

**Figure 3 ijms-26-04319-f003:**
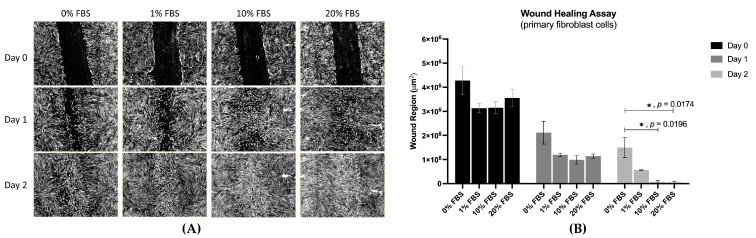
Wound closure in fibroblast cells treated with four specified concentrations of FBS (positive control) on days 0, 1, and 2. (**A**) Images from wound healing assays. (**B**) Effects of 0%, 1%, 10%, and 20% FBS on the wound region of fibroblast cells on days 0, 1, and 2. (* *p* < 0.05).

**Figure 4 ijms-26-04319-f004:**
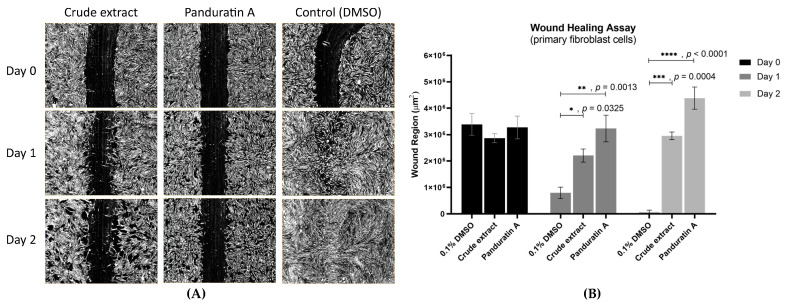
Wound closure in fibroblast cells treated with fingerroot extract, panduratin A, and DMSO on days 0, 1, and 2. (**A**) Images from wound healing assays. (**B**) Effects of 0.1% DMSO, fingerroot extract, and panduratin A on the wound region of fibroblast cells on days 0, 1, and 2. (* *p* < 0.05; ** *p* < 0.01; *** *p* < 0.001, **** *p* < 0.0001).

**Figure 5 ijms-26-04319-f005:**
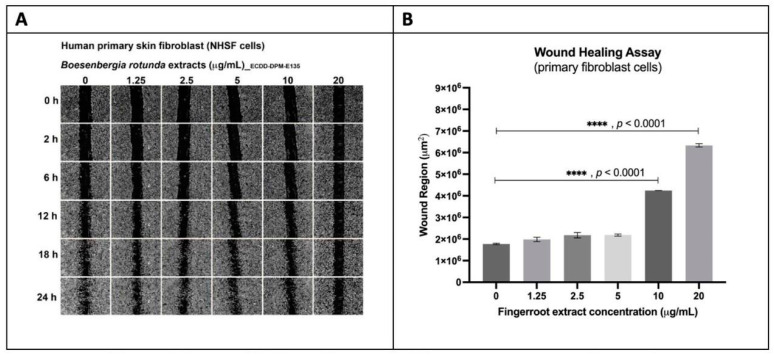
Wound closure in fibroblast cells treated with fingerroot extract over 24 h with six specified concentrations. (**A**) Images from wound healing assay. (**B**) Effects of 0, 1.25, 2.5, 5, 10, and 20 µg/mL fingerroot extract on the wound region of fibroblast at 24 h. (**** *p* < 0.0001).

**Figure 6 ijms-26-04319-f006:**
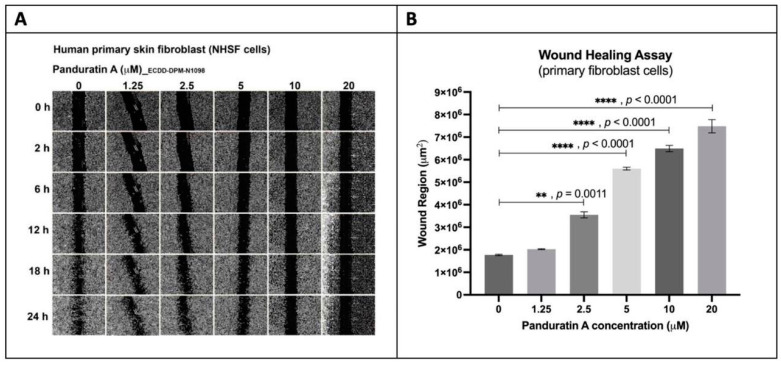
Wound closure in fibroblast cells treated with panduratin A over 24 h with six specified concentrations. (**A**) Images from wound healing assays. (**B**) Effects of 0, 1.25, 2.5, 5, 10, and 20 µg/mL panduratin A on the wound region of fibroblasts at 24 h. (** *p* < 0.01; **** *p* < 0.0001).

## Data Availability

The datasets used and analyzed in this study are available from the corresponding author on reasonable request.

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
