# Peer review of "Potential Cutaneous Applications of Boesenbergia rotunda Extract Based on Its In Vitro Anti-Melanogenic and Anti-Fibroproliferative Properties"

_ijms, 2025, doi:10.3390/ijms26094319_

Round 1

Reviewer 1 Report

Comments and Suggestions for Authors

  1. The manuscript contains grammatical typographical inconsistencies and awkward phrasing that impact clarity. For instance: Typographical: Merdicine” → “Medicine” in affiliations.
  2. In abstract section: Clearly structured but could be tightened for better flow, like “...was to investigate the anti-melanogenic and anti-tyrosinase effects...”
  • suggestion: “...was to evaluate the anti-melanogenic effects, including tyrosinase inhibition...
    1. In introduction section: Example: Lines 45–50 — provide more mechanistic insight into how UVA contributes to melanogenesis or fibroblast activation.
    2. Consider adding a short statement on the novelty of using UVA specifically.
    3. In the material and methods section: e.g., “was used to solubilize the purple formazan” → “to solubilize the formazan crystals”.
    4. Add one brief sentence on methodology (e.g., “using UVA-induced models in B16F10 melanoma cells and scratch assays in human fibroblasts”).
    5. UVA source specs (wavelength range of xenon arc lamp).
    6. Protein quantification method used before calculating tyrosinase and melanin content per mg protein.
    7. In results section: Wherever possible, include actual p-values instead of just thresholds.

Author Response

Response to Reviewer 1

We thank Reviewer 1 for their constructive feedback and careful reading of our manuscript. Below, we have addressed each comment in detail and revised the manuscript accordingly to improve clarity, scientific rigor, and overall quality.

Comment 1: The manuscript contains grammatical typographical inconsistencies and awkward phrasing that impact clarity. For instance: Typographical: “Merdicine” → “Medicine” in affiliations.

Response: Thank you for pointing this out. We have corrected the typographical error in the affiliations section and have thoroughly proofread the manuscript to address grammatical inconsistencies and improve clarity throughout. Mention exactly where in the revised manuscript this change can be found – page number 1, line 12.

Comment 2: In abstract section: Clearly structured but could be tightened for better flow, like “...was to investigate the anti-melanogenic and anti-tyrosinase effects...”
Suggestion: “...was to evaluate the anti-melanogenic effects, including tyrosinase inhibition...”

Response: We appreciate the suggestion. The abstract has been revised. The sentence now reads:
“The aim of this study was to investigate the anti-melanogenic including anti-tyrosinase effects of B. rotunda extract and panduratin A in B16F10 melanoma cells induced by UVA radiation.” Mention exactly where in the revised manuscript this change can be found – page number 1, line 31-33.

Comment 3-4:

In introduction section: Example: Lines 45–50 — provide more mechanistic insight into how UVA contributes to melanogenesis or fibroblast activation.

Consider adding a short statement on the novelty of using UVA specifically.

Response: Thank you for this helpful suggestion. In the revised manuscript, we have expanded the Introduction to discuss how UVA-induced ROS can trigger melanogenesis:

Revised Introduction

UVA radiation, which penetrates deeper into the dermis than UVB, plays a significant role in photoaging and hyperpigmentation. While traditionally associated with immediate pigment darkening via melanin oxidation, recent studies suggest that UVA also contributes to de novo melanogenesis through oxidative stress-related mechanisms. UVA-induced ROS formation leads to redox imbalance and oxidative DNA damage in melanocytes, which in turn can activate melanogenic enzymes such as tyrosinase, independent of classical UVB-induced p53 signaling. In this context, the antioxidant transcription factor, nuclear factor E2-related factor 2 (Nrf2) plays a critical role in mitigating UVA-mediated oxidative stress and maintaining cellular homeostasis in melanocytes and keratinocytes.

The novelty of our study lies in specifically addressing UVA-induced melanogenesis, an area less well-characterized than UVB-mediated pigmentation pathways.

Mention exactly where in the revised manuscript this change can be found – page number 2, line 62-73.

Revised Discussion Section related to UVA-induced melanogenesis (as reviewer 2 also requested)

Panduratin A is a cyclohexenyl chalcone derivative, a structural characteristic that may underlie its strong inhibitory effect on tyrosinase activity. In addition to its antimelanogenic properties, panduratin A is known for its anti-inflammatory, antioxidant, and antimicrobial activities, which have attracted attention for potential therapeutic applications. Likewise, B. rotunda exhibits significant antioxidant and anti-inflammatory properties. Previous studies have shown that B. rotunda enhances the cellular antioxidant defense system while simultaneously suppressing lipid peroxidation and modulating key signaling pathways, including protein kinase B (Akt) and nuclear factor kappa-light-chain-enhancer of activated B cells (NF-κB). These actions suggest that the photoprotective effects of B. rotunda in UV-induced melanogenesis may be attributed, at least in part, to its antioxidant mechanisms

Moreover, dysregulation of the antioxidant transcription factor Nrf2 has been shown to exacerbate UVA-induced melanogenesis, highlighting the redox-sensitive nature of this process. This suggests that, although UVA may not initiate melanogenic gene expression through the classical UVB-mediated pathways, it can still promote melanogenesis by inducing oxidative stress and disrupting redox signaling.

These findings are consistent with growing evidence that UVA radiation can promote melanogenesis through oxidative stress mechanisms, independent of its immediate pigment-darkening effects. Studies have demonstrated that UVA exposure leads to the accumulation of reactive oxygen species (ROS), oxidative damage, and depletion of intracellular glutathione, all of which are associated with increased tyrosinase activity and melanin content in melanocytes and melanoma cells. Notably, knockdown of nuclear factor erythroid 2–related factor 2 (Nrf2), a master regulator of cellular antioxidant responses, was shown to exacerbate UVA-induced melanogenic effects, implicating oxidative stress—rather than direct DNA damage—as a principal mediator of UVA-driven pigmentation. Moreover, these redox-sensitive responses are modulated by upstream mitogen-activated protein kinases (MAPKs), including ERK, JNK, and p38, which regulate the nuclear translocation and activity of Nrf2.

Additionally, Esposito et all., 2022 reviewed that compared to UVB, UVA causes significantly less erythema but is more potent in triggering pigment darkening—both immediate and persistent—as well as delayed tanning, particularly in individuals with darker skin tones. Unlike UVB, UVA does not directly impact key skin biomolecules. Instead, it acts indirectly by transferring energy to chromophores, which then produce reactive species that cause oxidative stress. Moreover, long-wavelength UVA and visible light (VL) can work together to enhance skin pigmentation and erythema. During typical daily activities, people are most commonly exposed to UVA and VL. However, current commercial sunscreens do not offer complete protection in this spectrum.

Taken together, our findings suggest that B. rotunda extract and its bioactive compound, panduratin A, may help reduce UVA-induced melanogenesis, as demonstrated by decreased melanin content and tyrosinase activity in treated cells. While the detailed mechanisms remain to be fully explored, these results support the potential of B. rotunda as a natural candidate for preventing UVA-related skin pigmentation.

Mention exactly where in the revised manuscript this change can be found – page number 9-10, line 198-239.

Comment 5: In the material and methods section: e.g., “was used to solubilize the purple formazan” → “to solubilize the formazan crystals”.

Response: We have revised the sentence for clarity. It now reads:
“Then, 200 µL of DMSO was used to solubilize the formazan crystals in each well.” Mention exactly where in the revised manuscript this change can be found – page number 11, line 308-309.

Comment 6: Add one brief sentence on methodology (e.g., “using UVA-induced models in B16F10 melanoma cells and scratch assays in human fibroblasts”).

Response: Thank you for this suggestion. We have added the following sentence to the Materials and Methods overview:
“The study utilized UVA-induced models in B16F10 melanoma cells to assess melanogenesis and scratch wound healing assays in human dermal fibroblasts to evaluate cellular migration.” Mention exactly where in the revised manuscript this change can be found – page number 11, line 280-282.

Comment 7: UVA source specs (wavelength range of xenon arc lamp).

Response: We have included the specifications of the UVA source in the Materials and Methods section. The revised text reads:
“After treatment with phenolics, cells were exposed to 8 J/cm2 UVA radiation (xenon arc lamp, with a wavelength range of 320–400 nm, Dermalight ultrA1).” Mention exactly where in the revised manuscript this change can be found – page number 12, line 315-317.

Comment 8: Protein quantification method used before calculating tyrosinase and melanin content per mg protein.

Response: We have added this information. The revised section states:
“Protein concentrations were determined using the Bradford assay, which allowed for accurate quantification of total protein in each sample. These values were then used to normalize tyrosinase activity and melanin content, ensuring that results were expressed relative to protein content (mg) for consistency across samples.” Mention exactly where in the revised manuscript this change can be found – page number 12, line 353-357.

Comment 9: In results section: Wherever possible, include actual p-values instead of just thresholds.

Response: We have revised the Results section to include actual p-values where available. For instance, “p < 0.05” is now stated as “p = 0.0339” (example), and all statistical data have been updated accordingly in figures. (Figure 2), (Results 2.2, 2.3)

Once again, we thank the reviewer for the insightful comments, which have greatly contributed to strengthening our manuscript.

Best regards,
Wilai Thanasarnaksorn, on behalf of all authors

Reviewer 2 Report

Comments and Suggestions for Authors

This manuscript investigates the effects of Boesenbergia rotunda and its bioactive compound, panduratin A, on melanogenesis and fibroblast proliferation. The figures are clear; however, several previously published studies have already presented similar findings. Moreover, this manuscript primarily demonstrates phenotypic outcomes without uncovering any novel underlying mechanisms. Therefore, I do not believe it is suitable for publication in its current form. Below are several specific comments:

  1. The title should be reconsidered. This study is limited to in vitro experiments, using a murine melanocyte cell line and human fibroblasts. Without in vivo validation, such as animal studies or assessments of safety, efficacy, and stability, it is premature to conclude that this compound is suitable for use as a cutaneous medication.
  2. UVA vs. UVB in Melanogenesis. 

    In the first set of results, melanogenesis and tyrosinase activity were evaluated under UVA irradiation. However, based on published literature, UVA and UVB exert different effects on melanocytes. UVB exposure induces tyrosinase gene expression and increases melanin biosynthesis over time. In contrast, UVA tends to cause immediate pigment darkening through oxidation of existing melanin but does not significantly induce melanogenesis-related gene expression. Please reconfirm your findings or include a discussion addressing this distinction.

    Reference:
    Choi W, Miyamura Y, Wolber R, Smuda C, Reinhold W, Liu H, Kolbe L, Hearing VJ. Regulation of human skin pigmentation in situ by repetitive UV exposure: molecular characterization of responses to UVA and/or UVB. J Invest Dermatol. 2010.

  3. Missing Citations. 

    Several important studies relevant to this topic were not cited. For example:

    a. Shim JS, Kwon YY, Han YS, Hwang JK. Inhibitory effect of panduratin A on UV-induced activation of mitogen-activated protein kinases (MAPKs) in dermal fibroblast cells. Planta Med. 2008.
    b. Lee CW, Kim HS, Kim HK, Kim JW, Yoon JH, Cho Y, Hwang JK. Inhibitory effect of panduratin A isolated from Kaempferia pandurata Roxb. on melanin biosynthesis. Phytother Res. 2010.

    These studies already demonstrated the role of panduratin A in fibroblast proliferation and melanin synthesis, but not mentioned in this manuscript.

    Additionally, in the “Discussion” section (lines 283–286 and 301–312), more citations are needed to support the claims.
  4. Fibroblast proliferation model.  In the second part of the study, fibroblast proliferation was tested using FBS as a positive control, DMSO as a negative control, and B. rotunda extract and panduratin A as experimental treatments. However, all wound healing experiments were conducted under normal culture conditions. To better mimic a keloid hypertrophic scarring environment, the addition of pro-fibrotic cytokines (TGF-β1, IL-6 or PDGF) should be considered to replicate the pathological state.
  5. If B. rotunda extract and panduratin A are shown to inhibit fibroblast proliferation under normal conditions, there is a potential concern that they might delay normal wound healing. This should be discussed in the manuscript.

Author Response

Response to Reviewer 2

We thank Reviewer for the detailed evaluation of our manuscript and for the thoughtful suggestions aimed at improving its scientific quality and clarity. Below, we provide a point-by-point response to each comment and describe the revisions made to the manuscript.

Comment 1: Title – In vitro limitation

The title should be reconsidered. This study is limited to in vitro experiments, using a murine melanocyte cell line and human fibroblasts. Without in vivo validation, such as animal studies or assessments of safety, efficacy, and stability, it is premature to conclude that this compound is suitable for use as a cutaneous medication.

Response:
We thank the reviewer for this important observation. We fully acknowledge that the current version of the title may overstate the applicability of our findings by implying direct therapeutic potential. As the study is confined to in vitro experiments, including cytotoxicity testing, functional assays on melanogenesis, and fibroblast proliferation using B16F10 murine melanocytes and human dermal fibroblasts, respectively, we have revised the title to more accurately reflect the scope and limitations of the research.

The updated title is: “In vitro evaluation of the anti-melanogenic and anti-fibroproliferative effects of Boesenbergia rotunda extract for potential dermatological applications”

We hope this revision clarifies our intentions and aligns with the reviewer's concerns regarding the manuscript’s scope.

Comment 2: UVA vs. UVB in Melanogenesis

UVA vs. UVB in Melanogenesis. 

In the first set of results, melanogenesis and tyrosinase activity were evaluated under UVA irradiation. However, based on published literature, UVA and UVB exert different effects on melanocytes. UVB exposure induces tyrosinase gene expression and increases melanin biosynthesis over time. In contrast, UVA tends to cause immediate pigment darkening through oxidation of existing melanin but does not significantly induce melanogenesis-related gene expression. Please reconfirm your findings or include a discussion addressing this distinction.

Reference:
Choi W, Miyamura Y, Wolber R, Smuda C, Reinhold W, Liu H, Kolbe L, Hearing VJ. Regulation of human skin pigmentation in situ by repetitive UV exposure: molecular characterization of responses to UVA and/or UVB. J Invest Dermatol. 2010.

Response:
We thank the reviewer for this important comment. Indeed, UVA and UVB trigger distinct pathways in skin pigmentation. While UVB is classically associated with delayed tanning through transcriptional upregulation of tyrosinase, increasing evidence suggests that UVA can also stimulate melanogenesis, particularly via oxidative stress-mediated mechanisms.

Revised Introduction

UVA radiation, which penetrates deeper into the dermis than UVB, plays a significant role in photoaging and hyperpigmentation. While traditionally associated with immediate pigment darkening via melanin oxidation, recent studies suggest that UVA also contributes to de novo melanogenesis through oxidative stress-related mechanisms. UVA-induced ROS formation leads to redox imbalance and oxidative DNA damage in melanocytes, which in turn can activate melanogenic enzymes such as tyrosinase, independent of classical UVB-induced p53 signaling. In this context, the antioxidant transcription factor, nuclear factor E2-related factor 2 (Nrf2) plays a critical role in mitigating UVA-mediated oxidative stress and maintaining cellular homeostasis in melanocytes and keratinocytes.

The novelty of our study lies in specifically addressing UVA-induced melanogenesis, an area less well-characterized than UVB-mediated pigmentation pathways.

Mention exactly where in the revised manuscript this change can be found – page number 2, line 62-73.

Revised Discussion Section related to UVA-induced melanogenesis (as reviewer 2 also requested)

Panduratin A is a cyclohexenyl chalcone derivative, a structural characteristic that may underlie its strong inhibitory effect on tyrosinase activity. In addition to its antimelanogenic properties, panduratin A is known for its anti-inflammatory, antioxidant, and antimicrobial activities, which have attracted attention for potential therapeutic applications. Likewise, B. rotunda exhibits significant antioxidant and anti-inflammatory properties. Previous studies have shown that B. rotunda enhances the cellular antioxidant defense system while simultaneously suppressing lipid peroxidation and modulating key signaling pathways, including protein kinase B (Akt) and nuclear factor kappa-light-chain-enhancer of activated B cells (NF-κB). These actions suggest that the photoprotective effects of B. rotunda in UV-induced melanogenesis may be attributed, at least in part, to its antioxidant mechanisms

Moreover, dysregulation of the antioxidant transcription factor Nrf2 has been shown to exacerbate UVA-induced melanogenesis, highlighting the redox-sensitive nature of this process. This suggests that, although UVA may not initiate melanogenic gene expression through the classical UVB-mediated pathways, it can still promote melanogenesis by inducing oxidative stress and disrupting redox signaling.

These findings are consistent with growing evidence that UVA radiation can promote melanogenesis through oxidative stress mechanisms, independent of its immediate pigment-darkening effects. Studies have demonstrated that UVA exposure leads to the accumulation of reactive oxygen species (ROS), oxidative damage, and depletion of intracellular glutathione, all of which are associated with increased tyrosinase activity and melanin content in melanocytes and melanoma cells. Notably, knockdown of nuclear factor erythroid 2–related factor 2 (Nrf2), a master regulator of cellular antioxidant responses, was shown to exacerbate UVA-induced melanogenic effects, implicating oxidative stress—rather than direct DNA damage—as a principal mediator of UVA-driven pigmentation. Moreover, these redox-sensitive responses are modulated by upstream mitogen-activated protein kinases (MAPKs), including ERK, JNK, and p38, which regulate the nuclear translocation and activity of Nrf2.

Additionally, Esposito et all., 2022 reviewed that compared to UVB, UVA causes significantly less erythema but is more potent in triggering pigment darkening—both immediate and persistent—as well as delayed tanning, particularly in individuals with darker skin tones. Unlike UVB, UVA does not directly impact key skin biomolecules. Instead, it acts indirectly by transferring energy to chromophores, which then produce reactive species that cause oxidative stress. Moreover, long-wavelength UVA and visible light (VL) can work together to enhance skin pigmentation and erythema. During typical daily activities, people are most commonly exposed to UVA and VL. However, current commercial sunscreens do not offer complete protection in this spectrum.

Taken together, our findings suggest that B. rotunda extract and its bioactive compound, panduratin A, may help reduce UVA-induced melanogenesis, as demonstrated by decreased melanin content and tyrosinase activity in treated cells. While the detailed mechanisms remain to be fully explored, these results support the potential of B. rotunda as a natural candidate for preventing UVA-related skin pigmentation.

Mention exactly where in the revised manuscript this change can be found – page number 9-10, line 198-239.

Comment 3:

Missing Citations. 

Several important studies relevant to this topic were not cited. For example:

  1. Shim JS, Kwon YY, Han YS, Hwang JK. Inhibitory effect of panduratin A on UV-induced activation of mitogen-activated protein kinases (MAPKs) in dermal fibroblast cells.Planta Med. 2008.
    b. Lee CW, Kim HS, Kim HK, Kim JW, Yoon JH, Cho Y, Hwang JK. Inhibitory effect of panduratin A isolated from Kaempferia pandurata Roxb. on melanin biosynthesis.Phytother Res. 2010.

These studies already demonstrated the role of panduratin A in fibroblast proliferation and melanin synthesis, but not mentioned in this manuscript.

Additionally, in the “Discussion” section (lines 283–286 and 301–312), more citations are needed to support the claims.

Response:
Thank you for bringing this to our attention. We have now cited many new references in the Introduction and Discussion sections:

Comment 4:

Fibroblast proliferation model.  In the second part of the study, fibroblast proliferation was tested using FBS as a positive control, DMSO as a negative control, and B. rotunda extract and panduratin A as experimental treatments. However, all wound healing experiments were conducted under normal culture conditions. To better mimic a keloid hypertrophic scarring environment, the addition of pro-fibrotic cytokines (TGF-β1, IL-6 or PDGF) should be considered to replicate the pathological state.

Response:
We acknowledge the reviewer's valid point regarding the limitations of our wound healing model. While this study aimed to assess the general proliferative and migratory response under normal culture conditions, we agree that the inclusion of pro-fibrotic stimuli would better simulate pathological wound environments.

As a result, we have now clearly stated this limitation in the Discussion:

“One limitation of our fibroblast model is the absence of pro-fibrotic cytokines such as TGF-β1, IL-6, or PDGF, which are typically elevated in keloid and hypertrophic scarring. Future studies incorporating these cytokines will be essential to determine whether B. rotunda and panduratin A selectively inhibit pathological over-proliferation without impairing normal wound healing.”

Mention exactly where in the revised manuscript this change can be found – page number 10-11, line 263-267.

Comment 5:

If B. rotunda extract and panduratin A are shown to inhibit fibroblast proliferation under normal conditions, there is a potential concern that they might delay normal wound healing. This should be discussed in the manuscript.

Response:
We agree that the potential for delayed wound healing due to anti-proliferative effects must be addressed. We have added the following cautionary note to the Discussion:

“Additionally, while reduced fibroblast proliferation may be beneficial in fibrotic conditions, it also raises potential concerns regarding delayed wound repair. Therefore, careful dose optimization and in vivo validation will be critical to distinguish the desired anti-fibrotic effects from unintended impairment of physiological healing processes.

Mention exactly where in the revised manuscript this change can be found – page number 11, line 267-271.

We once again thank Reviewer 2 for the critical and constructive feedback, which has helped us significantly strengthen the scientific quality and clarity of the manuscript.

Best regards,
Wilai Thanasarnaksorn, on behalf of all authors

Round 2

Reviewer 2 Report

Comments and Suggestions for Authors

I'm satisfied with the author's revisions. Thanks author's response, I have no further comments.